# Brain Extract of Subacute Traumatic Brain Injury Promotes the Neuronal Differentiation of Human Neural Stem Cells via Autophagy

**DOI:** 10.3390/jcm11102709

**Published:** 2022-05-11

**Authors:** Zhenghui He, Lijian Lang, Jiyuan Hui, Yuxiao Ma, Chun Yang, Weiji Weng, Jialin Huang, Xiongfei Zhao, Xiaoqi Zhang, Qian Liang, Jiyao Jiang, Junfeng Feng

**Affiliations:** 1Brain Injury Center, Department of Neurosurgery, Renji Hospital, School of Medicine, Shanghai Jiao Tong University, Shanghai 200127, China; hezhenghui8@sjtu.edu.cn (Z.H.); langlijian@sjtu.edu.cn (L.L.); 18424@renji.com (J.H.); mqsmyx@sjtu.edu.cn (Y.M.); yc_shsmu@sjtu.edu.cn (C.Y.); jiyaojiang@126.com (J.J.); 2Shanghai Institute of Head Trauma, Shanghai 200127, China; 19220@renji.com; 3Department of Biochemistry and Molecular Cell Biology, Shanghai Key Laboratory for Tumor Microenvironment and Inflammation, School of Medicine, Shanghai Jiao Tong University, Shanghai 200025, China; wengweiji@alumni.sjtu.edu.cn; 4Shanghai Angecon Biotechnology Co., Ltd., Shanghai 201318, China; zhaox@angecon.com (X.Z.); zhangxq@angecon.com (X.Z.); 5Department of Pathology, University of Texas Southwestern Medical Center, Dallas, TX 75390, USA; qian.liang@utsouthwestern.edu

**Keywords:** traumatic brain injury, neural stem cells, differentiation, autophagy

## Abstract

Background: After a traumatic brain injury (TBI), the cell environment is dramatically changed, which has various influences on grafted neural stem cells (NSCs). At present, these influences on NSCs have not been fully elucidated, which hinders the finding of an optimal timepoint for NSC transplantation. Methods: Brain extracts of TBI mice were used in vitro to simulate the different phase TBI influences on the differentiation of human NSCs. Protein profiles of brain extracts were analyzed. Neuronal differentiation and the activation of autophagy and the WNT/CTNNB pathway were detected after brain extract treatment. Results: Under subacute TBI brain extract conditions, the neuronal differentiation of hNSCs was significantly higher than that under acute brain extract conditions. The autophagy flux and WNT/CTNNB pathway were activated more highly within the subacute brain extract than in the acute brain extract. Autophagy activation by rapamycin could rescue the neuronal differentiation of hNSCs within acute TBI brain extract. Conclusions: The subacute phase around 7 days after TBI in mice could be a candidate timepoint to encourage more neuronal differentiation after transplantation. The autophagy flux played a critical role in regulating neuronal differentiation of hNSCs and could serve as a potential target to improve the efficacy of transplantation in the early phase.

## 1. Introduction

Traumatic brain injury (TBI) imposes an immense social burden [1,2,3,4]. The histopathological changes of TBI involve the damage of nervous and vascular tissue [5,6,7]. Thus, reconstructing the injured neuron circuit through grafting neural stem cells (NSCs) has been considered aa a promising cell therapy for TBI [8,9]. In a rodent TBI model, NSC transplantation has shown an improvement of cognitive and motor functions [10,11,12]. Recent studies also applied various growth factors, e.g., BDNF, GDNF, and bFGF, to promote the viability of grafted NSCs under TBI conditions [13,14,15].

However, the environment of TBI lesions is responsive to insults. Various factors can be released leading to neuroinflammation, oxidative stress, and cytotoxic edema [16,17,18,19,20]. These pathological conditions could influence grafted stem cells, including NSCs and mesenchymal stem cells. Previous studies reported that mouse mesenchymal stem cells pretreated with TBI brain extract exhibited a secretome that amplified neurogenesis [21] and differentiation to neurons [22]. The influence of the TBI environment on NSCs, however, has received less attention. These previous studies all used acute TBI brain extract. As the environment of TBI constantly changes, the influences of different phases after TBI on NSCs remain unknown.

To elucidate the influence of the TBI environment on NSCs, this study simulated an in vitro TBI environment for NSC culture by coculturing with TBI mouse brain extracts. We analyzed the neuronal differentiation of human NSCs in the acute/subacute/chronic phases of a mouse brain TBI extract, and explored the mechanism behind the influence of a TBI brain extract, to provide more understanding of TBI conditions and seek an early transplant timepoint of cell therapy for TBI.

## 2. Materials and Methods

### 2.1. Controlled Cortical Impact (CCI) Model and Brain Extract

Experimental animal usage was approved by the Animal Ethics Committee of Shanghai Renji Hospital. To obtain a definite and highly reproducible cortical lesion, the CCI model was created from 6–8-week old male C57BL/6 mice [23,24]. Mice were anesthetized with 4% isoflurane and placed in a stereotactic frame. A circular skull flap in the center of lambda and bregma with a 4 mm diameter was removed. Severe TBI was then generated by a CCI impactor, with a depth of 2 mm and velocity of 3 m/sec, according to a previous protocol [24].

On 1-, 3-, 7-, and 14-days post-injury (DPI), the brain extract was collected from the mice (Figure 1A). The collecting procedure followed that of previous research [21]. After anesthesia, mice were systematically perfused with pre-cooled phosphate-buffered saline (PBS). Then, the brains were rapidly removed and placed on ice. The 5 mm × 5 mm cortical tissue around the injury site was obtained and weighed. For the control group, the corresponding cortical tissue of the normal brain was obtained. These collected tissues were then homogenized in Dulbecco’s Modified Eagle medium/nutrient mixture F-12 (DMEM/F12) with a concentration of 150 mg/mL. The homogenate was centrifuged for 10 min at 10,000 rpm at 4 °C. The supernatant was then collected as the brain extract and stored at −80 °C.

### 2.2. Human NSCs (hNSCs) Culture and Treatment of Brain Extract

The human-embryo-derived neural stem cells were obtained from Shanghai Angecon Biotechnology Co., Ltd. For proliferation, the hNSCs were cultured in proliferate medium (DMEM/F12 + 2% B27 supplement + 1% N2 supplement + 20 ng/mL basic FGF + 20 ng/mL EGF + 1% Penicillin-Streptomycin). The basic FGF and EGF were removed to induce differentiation.

According to previous studies of the CCI mouse model, we identified the 1- and 3-DPI as the acute phase, the 7-DPI as the subacute phase, and the 14-DPI as the chronic phase. To identify the influence of different TBI brain extracts on hNSCs, cells were treated in a differentiate medium in addition to 10% corresponding brain extract for 24 h (Figure 1A). After that, the brain extracts were retracted and hNSCs were allowed to differentiate for 7 days.

For drug treatment, 3-Methyladenine (3-MA) (2 mM; Selleck, Shanghai, China) or Rapamycin (50 nM; Selleck, Shanghai, China) were added to the culture medium from the first day of hNSC differentiation and maintained for 24 h. After that, the medium was changed to the normal differentiate medium and hNSCs were left to differentiate for 7 days.

### 2.3. Protein Profile Analysis of Brain Extract

Proteomic analysis was performed by Shanghai Biotree biomedical technology Co., Ltd. The protein profiles of the brain extracts of 3-DPI, 7-DPI, and control groups were tested by TMT labeling, with a nano-UPLC (EASY-nLC1200) coupled to a Q Exactive HFX Orbitrap instrument (Thermo Fisher Scientific, Waltham, MA, USA). Each group had three independent samples.

### 2.4. Immunofluorescence

After 7-day differentiation, the hNSCs were fixed with 4% formaldehyde and incubated with anti-Human TUBB3 (1:300, #ab53623, Abcam, Cambridge, UK), anti-Human GFAP (1:300, #3670, Cell Signaling Technology, Danvers, MA, USA) or anti-Human SOX2 (1:300, #3579, Cell Signaling Technology, Danvers, MA, USA). Anti-Mouse IgG H&L (Alexa Fluor^®^ 594) (1:1000, #8890, Cell Signaling Technology, Danvers, MA, USA) and anti-rabbit IgG H&L (Alexa Fluor^®^ 488) (1:1000, #4412, Cell Signaling Technology, Danvers, MA, USA) were used as secondary antibodies. Nuclei were counterstained with DAPI stain (1:10,000; #4083, Cell Signaling Technology, Danvers, MA, USA). The slides were viewed using the Zeiss Axio observer Z1. The related mean fluorescence intensity of the TUBB3 to DAPI positive pixel was measured and calculated by ImageJ.

### 2.5. 5-Ethynyl-2-Deoxyuridine (EdU) Labeling

The proliferation of hNSCs in the injury environment was measured using an EdU assay (#C10310-3, Ribobio, Guangzhou, China). The hNSCs were treated in a corresponding brain extract in a differentiation medium for 24 h. After that, the EdU (50 μM) was added, and cells were incubated for another 24 h. The staining procedure followed the kit guideline. Images were obtained under fluorescence microscopy and analyzed by ImageJ.

### 2.6. Western Blot

Total proteins were extracted from the differentiated hNSCs using RIPA lysis buffer with 1 mM phenylmethanesulfonylfluoride. A total of 20 μg of protein samples was separated by a 10% SDS-PAGE gel and transferred onto polyvinylidene fluoride membranes. Each membrane was blocked for 1 h with 5% defatted milk powder at room temperature and then incubated with TUBB3 (1:1000, #ab53623, Abcam, Cambridge, UK), ATG5 (1:1000, #12994, Cell Signaling Technology, Danvers, MA, USA), BECN1 (1:1000, #3495, Cell Signaling Technology, Danvers, MA, USA), LC3A/B (1:1000, #12741, Cell Signaling Technology, Danvers, MA, USA), CTNNB (1:1000, #8480, Cell Signaling Technology, Danvers, MA, USA), MYC (1:1000, #5605, Cell Signaling Technology, Danvers, MA, USA) or ACTB (1:5000, #AB2001, ABways, Shanghai, China) primary antibody at 4 °C overnight. The membrane was then incubated with corresponding horseradish peroxidase-conjugated IgG secondary antibodies (1:10,000, #7074P2, Cell Signaling Technology, Danvers, MA, USA) for 1 h at room temperature. The blot images were captured by the gel Bio-rad imager. The gray value of each band was measured using ImageJ. The relative expression of each band was calculated by comparing the target protein band with GAPDH. Each group had three independent samples.

### 2.7. RNA Extraction and Quantitative Real-Time PCR Analysis

Total RNA was extracted from the differentiated hNSCs using an RNA-easy isolation reagent (#R701-01, Vazyme, Nanjing, China). The reverse transcription was performed by the HiScript II Q RT SuperMix for qPCR (#R233-01, Vazyme, Nanjing, China). Real-time PCR was performed with the LightCycler system (Roche, Boston, MA, USA) and ChamQ Universal SYBR qPCR Master Mix (#Q711-02, Vazyme, Nanjing, China) using specific primers. With GAPDH taken as an endogenous control, ATG5, BECN1, LC3, CTNNB, and MYC were detected. The primer sequences are shown in Table 1. The relative mRNA expression of each detected RNA was measured via ΔΔCt calculation. There were three samples in each group and each sample was tested three times.

### 2.8. Statistical Analyses

All data are presented as mean ± standard deviation (SD). One-way ANOVA was used for multiple comparisons using GraphPad software (Version 9.3.1). *p*-values < 0.05 were considered to be statistically significant.

## 3. Results

### 3.1. Human NSCs Differentiated a Higher Ratio of Neurons in Subacute/Chronic TBI Brain Extracts

The embryo-derived human neural stem cells (hNSCs) were well characterized (Appendix A). The brain extracts of TBI mice at 1-, 3-, 7-, and 14-DPI were collected and added into a differentiate medium to treat hNSCs for 24 h (Figure 1A). The mean fluorescence intensity (MFI) of TUBB3/DAPI increased gradually along with the increase in DPI (Figure 1B). The MFIs in the 7-DPI and 14-DPI groups were higher than in the 3-DPI group (Figure 1C). This MFI did not show a further increase between the 7-DPI and 14-DPI groups. To seek a candidate timepoint for early transplantation, the difference between 3-DPI and 7-DPI groups was surveyed in detail in the following research. The related protein and mRNA expression of TUBB3 in the 3-DPI (acute phase) and 7-DPI (subacute phase) groups further confirmed the higher neuronal differentiation in the subacute phase (Figure 1D,E). Taken together, these results indicate that the brain extract of 7-DPI was more favorable than that of the acute phases (1- and 3-DPI) for the neuronal differentiation of hNSCs. The differences between the acute and subacute phases could reveal the potential mechanism behind these influences.

### 3.2. Autophagy-Related Proteins Differentially Expressed in Acute and Subacute TBI Brain Extracts

To further reveal the different influences of the TBI brain extracts between the acute and subacute phases, protein profiling was employed, which detected a total of 6324 proteins in 6 samples of the acute and subacute phases. The different protein expression analysis identified 1318 differently expressed proteins in these 2 phases (Appendix A), including 969 up-regulated proteins and 349 down-regulated proteins in the subacute brain extract (Figure 2A). Among the top 20 up-regulated proteins in the subacute phase, seven proteins, Tnnt3, Sdhc, Cyc1, Rufy4, Gpnmb, Lactb, and Gpt2, were related to the activation of autophagy. Five autophagy-related proteins, Slc4a1, Lztr1, Apoc1, Hp, and Thbs1, were detected to be among the top 20 up-regulated proteins in the acute phase. The Z-score also showed a group of highly expressed proteins in the subacute TBI brain extract, including Vdac1, Mapt, Lrp1, Pdia3, and Aco2, which were also involved in autophagy (Figure 2B). The results of protein profiling highlight differentially activated autophagy in the acute and subacute phases after TBI, which might also trigger autophagy in hNSCs under the corresponding brain extract coculture.

### 3.3. Subacute TBI Brain Extract Promoted Neuronal Differentiation through the Activation of Autophagy in hNSCs

Autophagy plays an essential role in cells when encountering pathological conditions [25]. It was recently reported that autophagy participates in the differentiation of NSCs [26]. To understand the involvement of autophagy in hNSCs under different TBI brain extract cocultures, autophagy-related markers, ATG5, LC3A/B, and BECN1, were detected. As shown in Figure 3, hNSCs differentiated in the subacute TBI brain extract had significantly higher levels of ATG5, LC3A/B, and BECN1 than those in the acute group, which indicated a higher activation of autophagy of hNSCs in the subacute TBI brain extract. The activation of autophagy of hNSCs did not show a significant difference between the subacute and normal (control) groups. The activation of the WNT/CTNNB pathway, a reported coactivating signal pathway of autophagy, was also detected. The results showed that, in the acute TBI brain extract, the expression of CTNNB and MYC in hNSCs was insufficient. Further, hNSCs differentiated in the subacute and control groups had higher activated WNT/CTNNB pathways. These results together reveal that autophagy flux and the WNT/CTNNB pathway were involved and might play essential roles in regulating the neuronal differentiation of hNSCs within different the TBI brain extracts.

### 3.4. Activating Autophagy Rescued the Neuronal Differentiation within Acute TBI Brain Extract

To further confirm the role of autophagy in neuronal differentiation under the brain extracts, rapamycin was used to activate the autophagy of hNSCs in the acute group, and 3-methyladenine (3-MA) was applied to suppress autophagy in the subacute group [25,27]. 3-MA successfully suppressed Atg5, LC3AB, and Beclin-1 in hNSCs within the subacute TBI brain extract (Figure 4). The activation of the WNT pathway was also impaired with the treatment of 3-MA (Figure 4). The MFI of TUBB3 was also lower in the subacute+3-MA group, similar to the acute group (Figure 5). Furthermore, rapamycin activated autophagy flux in hNSCs cultured under the acute TBI brain extract conditions, which did not reach the activation level in the subacute group (Figure 4). The hNSCs treated with rapamycin also showed an activation of the WNT/CTNNB pathway (Figure 4). The immunofluorescence and detection of TUBB3 showed that, with rapamycin, the hNSCs differentiated more into neurons within the acute TBI brain extract, which was similar to the results from the subacute TBI brain extract (Figure 5). Taken together, it was confirmed that triggering autophagy could increase the neuronal differentiation of hNSCs. Higher activation in the subacute phase than in the acute phase was critical for the differentiation of hNSCs.

## 4. Discussion

In this study, we differentiated and cocultured hNSCs with TBI mouse brain extracts of different phases. We found that human NSCs cocultured with subacute and chronic TBI mouse brain extracts differentiated more into neurons than those with acute TBI mouse brain extracts. We also found that, under the influence of TBI brain extracts, the activation of autophagy and the WNT/CTNNB pathway played important roles in the neuronal differentiation of hNSCs (Figure 6). Further, the unfavorable influence of the acute phase TBI brain extracts could be reduced by activating autophagy through a rapamycin treatment.

Endogenous neurogenesis is limited due to the challenging environment after TBI [28,29]. Recently, NSCs have been successfully transplanted into TBI model rodents and shown to result in a significant improvement in neurological function [10,11,12]; however, clinical translation is still lacking. With TBI as a major cause of emergencies, different phases of TBI are drastically varying microenvironments to grafted NSCs. The influences of TBI on grafted NSCs from the acute phase to the chronic phase have long remained unknown, which have made an optimal transplant timepoint obscure.

In this study, we discussed this problem through coculturing hNSCs with TBI brain extracts of different DPIs. Our results found that, after the 7-DPI, the rate of neuronal differentiation did not increase further (Figure 1). Thus, the 7-DPI could be a candidate timepoint for NSC transplantation in mouse TBI. Moreover, we also found that cell proliferation was stimulated in the earlier stage after CCI, but this decreased as time progressed (Appendix A). This may imply that, for the acute phase, the transplantation of NSCs might largely differentiate into glial cells, while for the chronic phase, the inadequate stimulation of proliferation may not benefit a better reconstruction. Taken together, the cross point of decreased proliferation and increased differentiation (around 7 days post-injury, the subacute phase of mouse TBI) would be a balanced timepoint for early hNSCs transplantation.

The protein profile of TBI brain extracts revealed the microenvironment changes between the acute phase and subacute phase (Appendix A). Differential expressed proteins identified in GO classification reflected the cytokine reaction of TBI in brain tissue and were consistent with previous findings. The KEGG pathway analysis also revealed that metabolism-related processes and pathways were influenced in different phases (Appendix A). These findings were consistent with previous research [30,31,32,33,34]. According to previous reports, these changes in metabolism and the oxidative stress environment could activate an adaptive response such as autophagy in NSCs [35,36]. Moreover, autophagy-related proteins, e.g., ATG7, were also highly expressed in the subacute phase. Along with other autophagy-related proteins highly expressed in the two phases (Figure 2), these findings indicate that the activation of autophagy might participate in the response of hNSCs under different TBI microenvironments.

Generally, the autophagy pathway has been considered to have a critical role in modulating inflammatory response, oxidative stress, and cell apoptosis [25]. We further explored the activation of autophagy in exogenous NSCs under TBI brain extracts and the effect of autophagy on the differentiation of hNSCs. We found that the activation of autophagy in hNSCs was higher in the subacute group than in the acute group. The activation of autophagy through rapamycin could promote the neuronal differentiation of NSCs under the acute TBI brain extract conditions. Previous studies showed that the involvement of autophagy was essential for the oligodendrocyte and astrocyte differentiation of NSCs [37,38]. Li et.al found that, in mice NSCs, the activation of autophagy could induce the NSCs to differentiate into immature neurons [27]. Combined with our results, these findings confirmed that autophagy plays a critical role in the differentiation of NSCs. Evidence of the downstream regulation of autophagy on neuronal differentiation is lacking; however, a considerable number of studies have reported that the activation of autophagy could initiate the WNT/CTNNB signaling pathway [39,40]. In this study, we found that the expression level of MYC, a reported downstream gene of CTNNB regulating NSC differentiation [41,42,43], was increased along with the activation of autophagy. Therefore, the autophagic flux and WNT/CTNNB composed an essential response mechanism of exogenous hNSCs within TBI brain extract simulation.

There were several limitations in this study. The species inconsistency between NSCs and the TBI models is a major concern. Future experiments using consistent species (mouse NSCs) should be considered. Human clinical samples, e.g., blood and cerebrospinal fluid, could also be used to reveal an optimal transplantation time window similar to the subacute phase in mice. In vivo studies should be completed in future research.

## 5. Conclusions

In vitro, hNSCs had an increased ratio of neuronal differentiation within mouse TBI brain extract conditions from the acute phase to the subacute phase, but entered a plateau phase after 7-DPI. The subacute phase in mouse TBI could be a candidate timepoint for early hNSCs transplantation. Moreover, autophagy flux played an essential role in the neuronal differentiation of hNSCs under TBI brain extract conditions, which provides potential intervention targets for transplantation research.

## Figures and Tables

**Figure 1 jcm-11-02709-f001:**
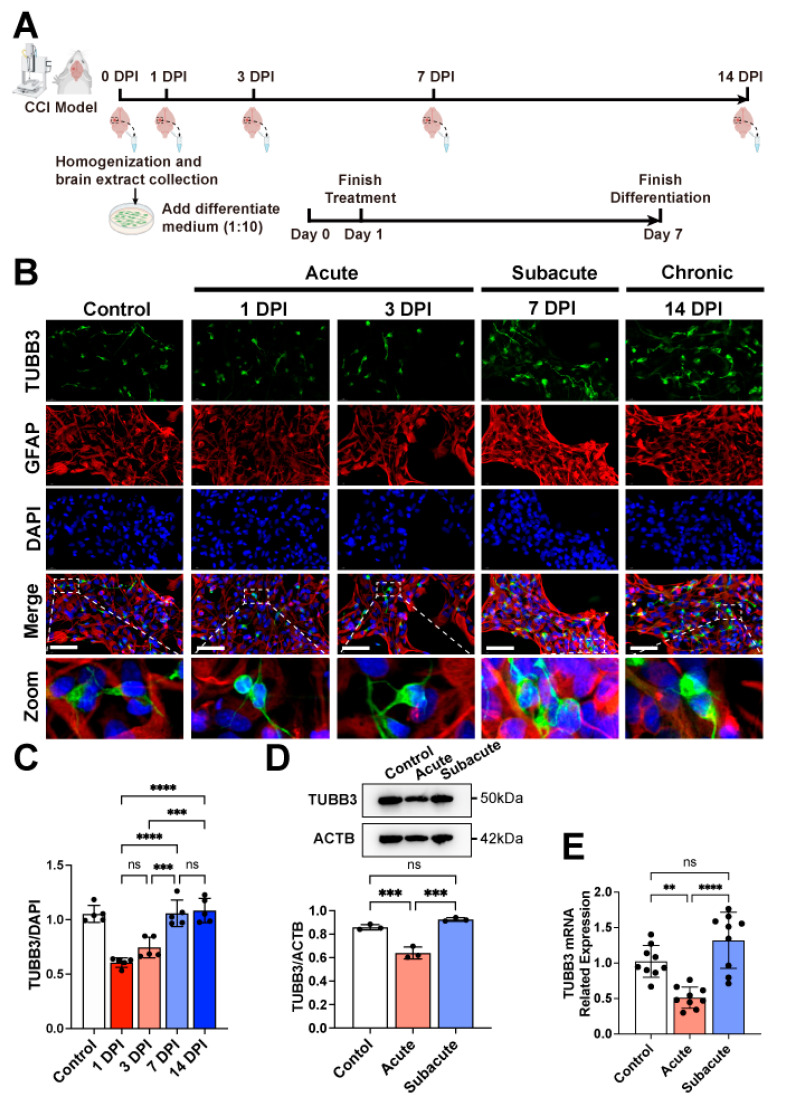
**The neuronal differentiation of hNSCs within TBI brain extract at different phases.** The brain extracts of CCI mice with different injury times were collected and added to a hNSC differentiate medium with a ratio of 1:10 to simulate the TBI microenvironment in vitro (**A**). Immunofluorescence revealed that hNSCs could differentiate into TUBB3^+^ neurons and GFAP^+^ astrocytes (**B**). The statistical analysis of mean fluorescence intensity (MFI) of TUBB3/DAPI showed an increasing ratio of TUBB3^+^ cells with the progression of days post-injury (**C**). Related protein and mRNA expression of TUBB3 also confirmed differences between the acute and subacute phases (**D**,**E**). The black circle in (**C**–**E**) represented the individual value of each sample. Bar scale: 50 μm; **: *p* < 0.01; ***: *p* < 0.001; ****: *p* < 0.0001; ns: *p* ≥ 0.05. CCI: Controlled Cortical Impact; DPI: Day Post-injury.

**Figure 2 jcm-11-02709-f002:**
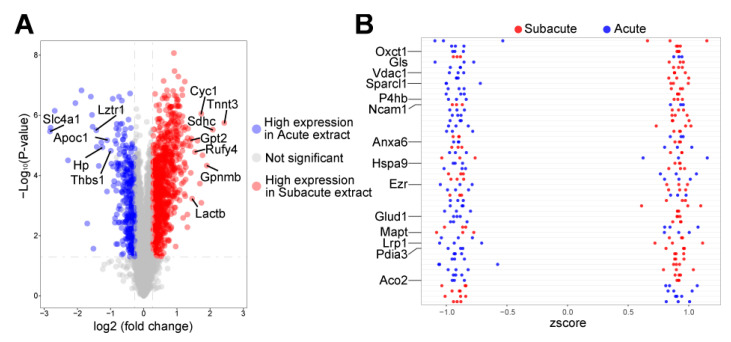
**Differently expressed proteins of the acute and subacute brain extracts.** Autophagy-related proteins were detected in the TBI microenvironment in the acute and subacute phases (**A**,**B**).

**Figure 3 jcm-11-02709-f003:**
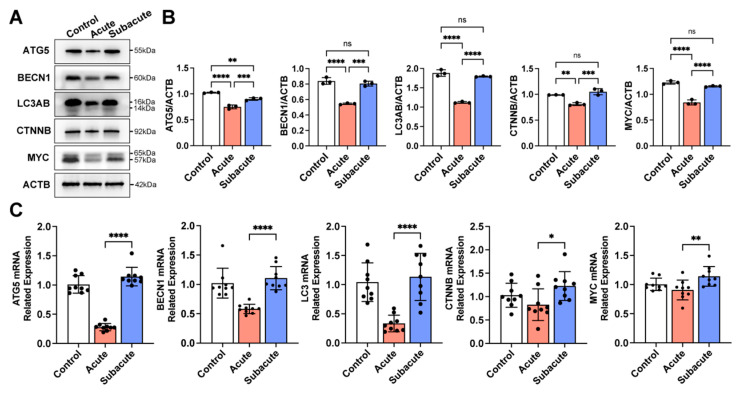
**The activation of autophagy flux and the WNT/CTNNB pathway in hNSCs within TBI brain extracts.** Markers of autophagy flux (ATG5, BECN1, and LC3AB) and the WNT/CTNNB pathway (CTNNB and MYC) were detected (**A**). Related expression of these proteins and mRNAs indicated that autophagy flux and the WNT/CTNNB pathway were strongly activated in the subacute TBI microenvironment (**B**,**C**). The black circle in B,C represented the individual value of each sample. *: *p* < 0.05; **: *p* < 0.01; ***: *p* < 0.001; ****: *p* < 0.0001; ns: *p* ≥ 0.05.

**Figure 4 jcm-11-02709-f004:**
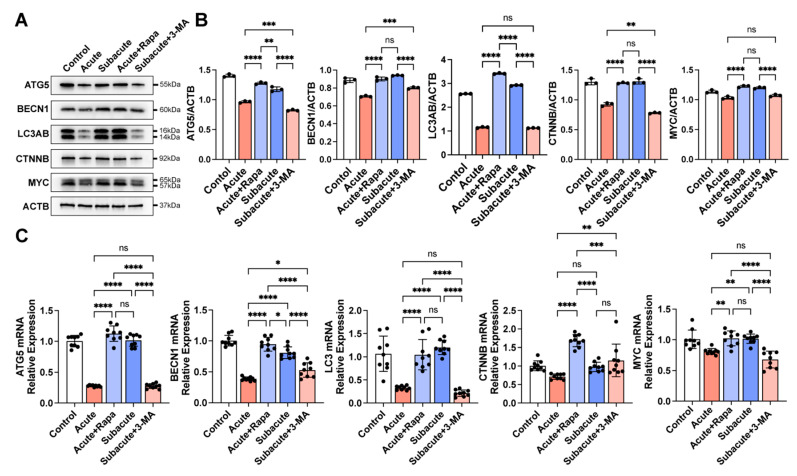
**Intervening autophagy flux in hNSCs in TBI brain extracts at different phases.** Markers of autophagy flux and the WNT/CTNNB pathway were detected (**A**). Related expression of proteins and mRNAs revealed that rapamycin (50 nM) could activate autophagy flux in the acute phase and 3-MA (2 μM) could inhibit autophagy flux in the subacute phase (**B**,**C**). The activation of the WNT/CTNNB pathway showed a corresponding change in the autophagy flux (**B**,**C**). The black circle in B,C represented the individual value of each sample. *: *p* < 0.05; **: *p* < 0.01; ***: *p* < 0.001; ****: *p* < 0.0001; ns: *p* ≥ 0.05.

**Figure 5 jcm-11-02709-f005:**
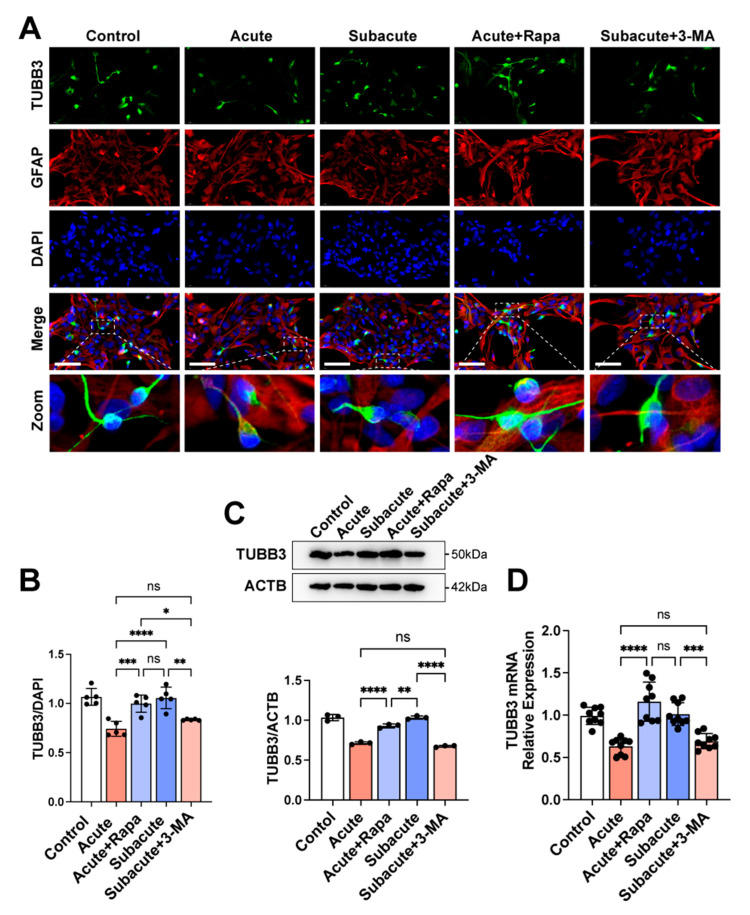
**Treating with rapamycin promoted the neuronal differentiation of hNSCs in the acute TBI brain extract.** With a treatment of rapamycin (50 nM) or 3-MA (2 μM), hNSCs could differentiate into TUBB3^+^ neurons and GFAP^+^ astrocytes in stimulated TBI microenvironments (**A**). The mean fluorescence intensity (MFI) of TUBB3/DAPI showed that the treatment with rapamycin increased neuronal differentiation in the acute phase, while treating with 3-MA in the subacute phase decreased neuronal differentiation (**B**). Related protein and mRNA expression of TUBB3 also indicated the same result (**C**,**D**). The black circle in B,C,D represented the individual value of each sample. Bar scale: 50 μm; *: *p* < 0.05; **: *p* < 0.01; ***: *p* < 0.001; ****: *p* < 0.0001; ns: *p* ≥ 0.05.

**Figure 6 jcm-11-02709-f006:**
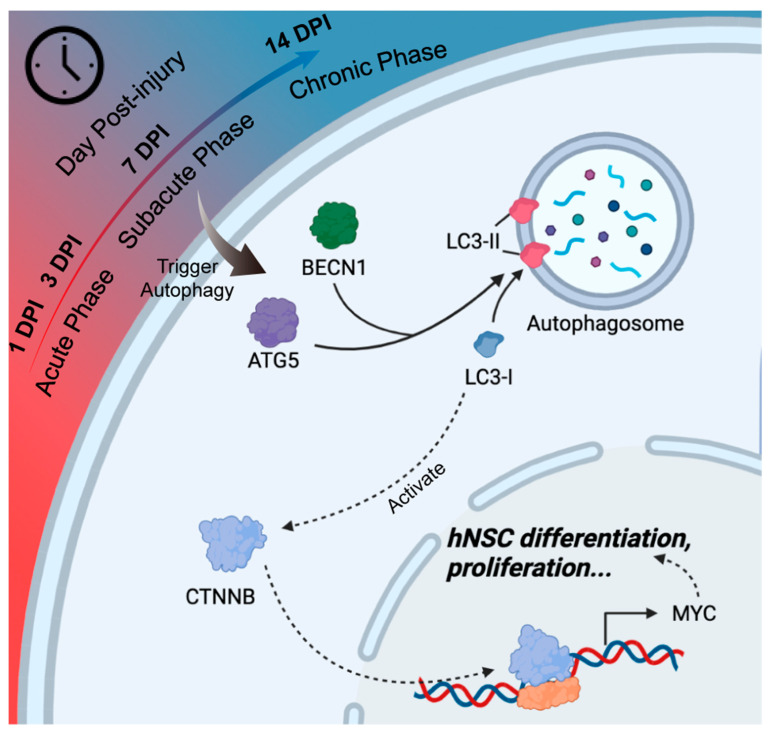
**Mechanism diagram of the influence of TBI brain extracts on the neuronal differentiation of hNSCs.** TBI brain extracts with different phases differently activated the autophagy flux of hNSCs. In hNSCs, the activation of the autophagy flux correlated with the WNT/CTNNB pathway, which boosted the differentiation and proliferation of hNSCs by regulating the transcription of MYC.

**Table 1 jcm-11-02709-t001:** Quantitative real-time polymerase chain reaction primer sequences.

Gene	Direction	Primer Sequence (5′-3′)
GAPDH	Forward	GGAGTCCACTGGCGTCTTCA
	Reverse	GTCATGAGTCCTTCCACGATACC
ACTB	Forward	CATGTACGTTGCTATCCAGGC
	Reverse	CTCCTTAATGTCACGCACGAT
ATG5	Forward	AGAAGCTGTTTCGTCCTGTGG
	Reverse	AGGTGTTTCCAACATTGGCTC
BECN1	Forward	GGTGTCTCTCGCAGATTCATC
	Reverse	TCAGTCTTCGGCTGAGGTTCT
LC3	Forward	AACATGAGCGAGTTGGTCAAG
	Reverse	GCTCGTAGATGTCCGCGAT
CTNNB	Forward	CATCTACACAGTTTGATGCTGCT
	Reverse	GCAGTTTTGTCAGTTCAGGGA
MYC	Forward	TCCCTCCACTCGGAAGGAC
	Reverse	CTGGTGCATTTTCGGTTGTTG
NES	Forward	CTGCTACCCTTGAGACACCTG
	Reverse	GGGCTCTGATCTCTGCATCTAC
SOX2	Forward	GCCGAGTGGAAACTTTTGTCG
	Reverse	GGCAGCGTGTACTTATCCTTCT
SOX1	Forward	CAGTACAGCCCCATCTCCAAC
	Reverse	GCGGGCAAGTACATGCTGA

## Data Availability

The data that support the findings of this study are available from the corresponding author upon reasonable request.

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
