# Peer review of "Brain Extract of Subacute Traumatic Brain Injury Promotes the Neuronal Differentiation of Human Neural Stem Cells via Autophagy"

_jcm, 2022, doi:10.3390/jcm11102709_

Round 1

Reviewer 1 Report

here is my Reviewer-Summary: after thorough and time-consuming revision, I do not recommend publishing this paper in the current shape. There are severe issues regarding wording, sentence formation, grammar etc. (examples given afterwards). The aim of the authors to show that hNSCs had an increased ratio of neuronal differentiation under the stimulation of mouse TBI brain extract from acute phase to subacute phase, and the conclusion that the subacute phase in mouse TBI could be a candidate timepoint for early hNSCs transplantation, could be shown, however, in a very unclear way, there are too many technical details which could be easily presented as supplementary information, but distract the reader from the main message.

ABSTRACT  - should be completely reworked

line 16: "after traumatic brain.... in brain would be dramatically changed" - instead of would be rather cell environment is dramatically changed

line 18: is instad of was

line 19: either a TBI mouse or TBI mice....

INTRODUCTION

line 34/35: unnecessary first sentence - "fatal silent epidemic" stagily wording

line 42: exchange: "et. al" in that context - etc., and so forth, and so on

line 51: "We furtherly researched the..." grammar issue

MATERIALS + METHODS - too much details, only technical parameters, so could be given in the supplementary materials - no novelties regarding the experimental set up and procedures

line 60: question: why are ther used only male mice? what differences would you expect in female mice and the hormonal influence regarding an autophagiy process or healing/revitalizing/differentiation of cells?

line 63: "a circle skull" - wrong wording

line 90, 98: the i is missing in bran - its supposed to be brain

line 158: "2.8 Statical analysis" - I assume it is supposed to be statistical analyses

line 164: there is some words/articles missing in the headline

line 169: with the stain of - grammar wrong

line 174/175: surveyed in following research - missing article: the/this

FIGURES:

all of the included figures were pixelated and blurred, so for me as a reader I could literally not see the details (even not when zooming in) and therefore it was impossible to grasp the message of the figures at all.

SUM:

I think the research approach has potential, however, the way the paper is written, procedures explained (or not explained: eg. where are the details regarding the mass spectrometry?) and results are deduced and discussed will need a complete new revised version.

Unfortunately the poor language skills make it additionally challenging for the reader to keep "on track" and be able subsequently to follow the thread. 

Author Response

Response to Reviewer 1 Comments

here is my Reviewer-Summary: after thorough and time-consuming revision, I do not recommend publishing this paper in the current shape. There are severe issues regarding wording, sentence formation, grammar etc. (examples given afterwards). The aim of the authors to show that hNSCs had an increased ratio of neuronal differentiation under the stimulation of mouse TBI brain extract from acute phase to subacute phase, and the conclusion that the subacute phase in mouse TBI could be a candidate timepoint for early hNSCs transplantation, could be shown, however, in a very unclear way, there are too many technical details which could be easily presented as supplementary information, but distract the reader from the main message.

Reply:

Thank you so much for your thorough and extremely careful review of our study.

We are very sorry the manuscript was not written well enough which lead to a confusing and time-consuming review. The manuscript has now been carefully revised especially in language. And all your important comments were replied point by point.

Again, we sincerely appreciate all your suggestions and help in improving the manuscript. Thank you.

ABSTRACT - should be completely reworked

Reply:

Thank you for your comment. The ABSTRACT has been rewritten to concisely display the aim of our study. All grammar issues have been carefully revised.

line 16: "after traumatic brain.... in brain would be dramatically changed" - instead of would be rather cell environment is dramatically changed

Reply:

Thank you for the suggestion. The corresponding wording has been revised.

line 18: is instad of was

Reply:

The corresponding wording has been revised. Thank you.

line 19: either a TBI mouse or TBI mice....

Reply:

Thank you for the suggestion. The corresponding wording has been revised.

INTRODUCTION

line 34/35: unnecessary first sentence - "fatal silent epidemic" stagily wording

Reply:

Thank you for this suggestion. The grammar issues in the Introduction section have been thoroughly revised.

line 42: exchange: "et. al" in that context - etc., and so forth, and so on

Reply:

Thank you for the grammar suggestion. The corresponding wording has been revised.

line 51: "We furtherly researched the..." grammar issue

Reply:

Thank you for the grammar suggestion. The corresponding wording has been revised.

MATERIALS + METHODS - too much details, only technical parameters, so could be given in the supplementary materials - no novelties regarding the experimental set up and procedures

Reply:

Thank you for this suggestion. This section has been revised to a more concise form.

line 60: question: why are ther used only male mice? what differences would you expect in female mice and the hormonal influence regarding an autophagiy process or healing/revitalizing/differentiation of cells?

Reply:

Thank you for raising this point. It has been reported that due to hormonal influences, male and female mice will exhibit different pathological responses to traumatic brain injury (TBI) 1. Preclinical studies all found that females had better physiological outcomes than male mice after TBI 2. A recent review also showed that sex differences played roles in autophagy-mediated diseases 3. For these reasons, we used only male mice to eliminate the effects of gender.

The below are references for our reply:

[1] Jullienne, Amandine et al. “Male and Female Mice Exhibit Divergent Responses of the Cortical Vasculature to Traumatic Brain Injury.” Journal of neurotrauma vol. 35,14 (2018): 1646-1658.

[2] Mollayeva, Tatyana et al. “Traumatic brain injury: sex, gender and intersecting vulnerabilities.” Nature reviews. Neurology vol. 14,12 (2018): 711-722.

[3] Shang, Dangtong et al. “Sex differences in autophagy-mediated diseases: toward precision medicine.” Autophagy vol. 17,5 (2021): 1065-1076.

line 63: "a circle skull" - wrong wording

Reply:

Thank you for the suggestion. The corresponding phrase has been revised.

line 90, 98: the i is missing in bran - its supposed to be brain

Reply:

Thank you for the suggestion. This has been revised.

line 158: "2.8 Statical analysis" - I assume it is supposed to be statistical analyses

Reply:

Thank you for the suggestion. This has been revised.

line 164: there is some words/articles missing in the headline

Reply:

Thank you for pointing it out. This has been revised.

line 169: with the stain of - grammar wrong

Reply:

Thank you for the grammar suggestion. The corresponding phrase has been revised.

line 174/175: surveyed in following research - missing article: the/this

Reply:

Thank you for the grammar suggestion. This has been revised.

FIGURES:

all of the included figures were pixelated and blurred, so for me as a reader I could literally not see the details (even not when zooming in) and therefore it was impossible to grasp the message of the figures at all.

Reply:

Thank you for raising this point, we feel sincerely sorry for this fault. We have enlarged all figures in the manuscript to ensure quality after transformation to PDF format. We also uploaded original figures (300 ppi) as supplementary files for your review.

SUM:

I think the research approach has potential, however, the way the paper is written, procedures explained (or not explained: eg. where are the details regarding the mass spectrometry?) and results are deduced and discussed will need a complete new revised version.

Unfortunately the poor language skills make it additionally challenging for the reader to keep "on track" and be able subsequently to follow the thread.

Reply:

Thank you for your positive comment on the research itself.

And we sincerely thank you for the comments on the weakness of the paper. We thus have completely revised our article following your suggestions, especially the Abstract, Methods, and Discussion sections. This revised article has also been edited by a native speaker to eliminate language mistakes (Ms Loren Skudder-Hill, from Department of Neurosurgery, Yuquan Hospital Affiliated to Tsinghua University School of Medicine, who is mentioned in the Acknowledgements).

Again, we highly appreciate your careful review and all the comments which improves this paper so much.

Reviewer 2 Report

This is a well-written paper containing interesting results which merit publication. They have provided evidences to demonstrate TBI brain extract with subacute phase intensely triggered the autophagy flux in hNSCs, leading to the activation of the WNT/CTNNB pathway. These results suggested that the subacute phase around 7 days after TBI in mice could be a candidate timepoint for early hNSCs transplantation. This is a very important issue in stem cell therapy.

Suggestion: To find the pathway involved in the different phase TBI influences on differentiation of human NSCs , the experiment could also be determined by phosphoproteomics or other bioinformatics (Ingenuity Pathway Analysis) to provide more information.

Author Response

Response to Reviewer 2 Comments

This is a well-written paper containing interesting results which merit publication. They have provided evidences to demonstrate TBI brain extract with subacute phase intensely triggered the autophagy flux in hNSCs, leading to the activation of the WNT/CTNNB pathway. These results suggested that the subacute phase around 7 days after TBI in mice could be a candidate timepoint for early hNSCs transplantation. This is a very important issue in stem cell therapy.

Suggestion: To find the pathway involved in the different phase TBI influences on differentiation of human NSCs, the experiment could also be determined by phosphoproteomics or other bioinformatics (Ingenuity Pathway Analysis) to provide more information.

Reply:

(1) Thank you for your approval of our study.

(2) We are also in great agreement with your suggestion on further analyses. As an in-vitro study, our protein profile of brain extracts revealed changes in the TBI microenvironment from acute to subacute phases. In the supplementary figures, GO and KEGG analyses also revealed related pathways involved in TBI influences. Your suggestion recommended more robust proteomic techniques to greater understand these changes. We believe this will be of great help in future in-vivo research.

Reviewer 3 Report

First of all, I would like to congratulate the authors for the excellent work.

They focused on the timing of neuronal cell transplantation in sTBI and found that the neuronal cell transplantation has the maximum effect of neuronal differentiation in subacute phase of sTBI. Furthermore, there was no significant difference after 7 days of sTBI. As pathophysiological mechanism, they used proteomic analysis to identify protein and mRNA up- and downregulation in different phases of TBI and found that autophagy regulation and WNT/CTNNB pathway are important mechanism for neuronal differentiation. It is of interest, if the result of the study can be translated to human beings and if the time course is similar to a mice.

Prior to publication, there is one minor issue. The figures have low quality which is difficult to read. Thus, the quality of those figures needed to be revised. 

Author Response

Response to Reviewer 3 Comments

First of all, I would like to congratulate the authors for the excellent work.

They focused on the timing of neuronal cell transplantation in sTBI and found that the neuronal cell transplantation has the maximum effect of neuronal differentiation in subacute phase of sTBI. Furthermore, there was no significant difference after 7 days of sTBI. As pathophysiological mechanism, they used proteomic analysis to identify protein and mRNA up- and downregulation in different phases of TBI and found that autophagy regulation and WNT/CTNNB pathway are important mechanism for neuronal differentiation. It is of interest, if the result of the study can be translated to human beings and if the time course is similar to a mice.

Prior to publication, there is one minor issue. The figures have low quality which is difficult to read. Thus, the quality of those figures needed to be revised.

Reply:

(1) Thank you for your praise of our study.

(2) We feel sincerely sorry for this mistake in figure quality. In the revised version, we have enlarged all figures in the manuscript to ensure high quality after transformation to PDF format. We also uploaded original figures (300ppi) as supplementary files for your review.
